# Asking the Experts: Using Cognitive Interview Techniques to Explore the Face Validity of the Mental Wellness Measure for Adolescents Living with HIV

**DOI:** 10.3390/ijerph20054061

**Published:** 2023-02-24

**Authors:** Zaida Orth, Brian Van Wyk

**Affiliations:** School of Public Health, Faculty of Community and Health Sciences, University of the Western Cape, Robert Sobukwe Rd., Bellville 7535, South Africa

**Keywords:** adolescents living with HIV, instrument development, mental wellness, cognitive interviewing

## Abstract

There has been an increased focus on the mental health of adolescents living with HIV (ALHIV), because evidence shows that poor mental health outcomes are associated with lower rates of adherence and retention in HIV care. However, the research to date has predominantly focused on addressing mental health problems and reducing symptoms of mental illness rather than strengthening mental wellness [positive mental health]. Consequently, little is known about the critical mental wellness indicators that should be targeted in services for ALHIV. There is a need for valid and appropriate measures of mental wellness to drive research and provide evidence on the mental wellness needs of ALHIV that would inform service delivery as well as the monitoring and evaluation of treatment outcomes. To this end, we developed the Mental Wellness Measure for Adolescents Living with HIV (MWM-ALHIV) for ALHIV in the South African context. In this paper, we report on the findings from a cognitive interview study with nine ALHIV aged 15–19 years receiving treatment at a public healthcare facility in the Cape Metropole, South Africa. Through interviews, participants identified key issues related to the wording, relevance and understanding of the items and provided suggestions to improve the instrument’s overall face validity.

## 1. Introduction

Mental wellness (positive mental health) has been identified as a significant driver of adolescent health and well-being [1]. This is evidenced in the inclusion of mental health (wellness) in the Sustainable Development Goal (SDG) target 3, which highlights mental health promotion as a critical factor in reducing premature mortality from non-communicable diseases and ensuring good health for all by 2030 [2]. This target is especially relevant for adolescents living with HIV (ALHIV), as they learn to navigate living with a highly stigmatized chronic condition and other challenges and health risks associated with adolescence [3]. Research aimed at understanding the lived experiences of ALHIV has shed light on the various biopsychosocial challenges face, including delayed disclosure, stigma, dysfunctional families, community violence, substance use, poverty, gender inequality and bullying, to name but a few [4,5,6,7]. Their exposure to these various stressors places them at greater risk of developing mental health disorders in comparison to their peers who do not have HIV [4,5,6,7]. Various studies have reported high prevalence rates of mental health disorders such as anxiety, depression, and post-traumatic stress disorder (PTSD) [7,8,9]. Moreover, poor mental health has been significantly associated with low adherence to antiretroviral therapy (ART) and retention in care (RiC) among ALHIV [10,11]. Consequently, the low rates of adherence to ART and RiC are associated with higher AIDS-related deaths among ALHIV in comparison to children and adults living with HIV [12,13].

South Africa is at the epicenter of the HIV pandemic, and is home to approximately 20% of the global adolescent HIV population aged 10–19 years [13]. The increased prevalence of ALHIV in South Africa is attributed to new infections as well as the successful roll-out of the ART program, which has increased the survival rate of perinatally inflected children [14]. In addition to navigating the physical, social and psychological changes experienced during adolescence and the challenges associated with managing HIV, ALHIV also have to navigate the complex myriad of socio-economic, systemic and environmental factors which shape their daily lives and impact their physical and mental health and wellbeing [14]. The current context of HIV treatment and care is rooted in the socio-political economic system from the apartheid era, which continues to perpetuate racial-based inequalities, thereby fostering conditions in which certain groups are more susceptible to HIV-infection [15]. For example, results from a 2012 population survey indicated that Black South African women were disproportionally affected by HIV in comparison to other groups [15].

Within the South African context, research has shown that structural factors such as poverty, housing, a lack of service provision in communities, racial inequalities, violence, and gender inequalities negatively impact the mental health of ALHIV [14]. For example, reports indicate that the majority of ALHIV belong to marginalized populations characterized by poverty. As a consequence, the impact of poverty is associated with limited access to resources, food insecurity and education, which in turn is associated with poor mental health outcomes and adherence [14]. Furthermore, a study by Woollet et al. [9] aimed at identifying the mental health risks among ALHIV accessing care at public healthcare facilities in Johannesburg found that there were high levels of reported mental health problems: 27% were symptomatic for depression, anxiety, or PTSD; and 24% reported suicidality. Additionally, the findings indicated that hunger, violence, gender, and illness were significantly correlated with mental health problems, while mental wellness factors such as hope for the future and knowledge of status were considered as protective factors [9]. These findings indicate that ALHIV require additional support to manage their condition in relation to the contextual risks that they experience. It is critical to address the structural problems and inequalities ALHIV face, yet solutions to these problems are complex and require transforming the macro contexts and political systems. However, preventative strategies that improve the mental wellness of ALHIV are useful in strengthening their capacities to be resilient and grow to become productive members of society. 

In support of mental wellness, studies evaluating the effectiveness of psychosocial interventions in improving mental health and adherence among ALHIV have shown promising results [16,17,18,19]. This evidence has increased calls and advocacy to improve mental health (wellness) promotion for ALHIV and to integrate mental health care into adolescent-friendly services [20,21]. However, many of the studies have focused on reducing mental health problems rather than promoting mental wellness. Quantitative studies that measure the effectiveness of psychosocial interventions in improving mental health among ALHIV often use instruments measuring symptoms of mental illness. Therefore, improvements to mental health in these studies are based on the measured reduction of mental illness symptoms. To truly establish the effectiveness of psychosocial interventions or mental health services in improving mental health among ALHIV, we need to develop holistic evaluations by measuring both mental health problems and mental wellness [21,22]. 

There has been a proliferation of positive psychology research that focused on a range of dimensions from theory development, evaluation of positive psychology interventions, and instrument testing [23]. However, despite the relevance of positive psychology as a tool to promote health and prevent illness (especially in the context of low-resourced settings), the majority of such research has been done in Western contexts, with only small pockets of evidence emerging in the global South [23]. Furthermore, despite the increased focus on positive mental health, a challenge of the field is the lack of consensus on how mental wellness should be conceptualized [1]. Research has identified a range of mental wellness constructs such as self-acceptance, hope, connectedness, and life satisfaction, among others, that are associated with improved well-being and positive functioning in adults and adolescents [1,24,25]. As mentioned, these concepts have mostly emerged from research in the Western context, with a number of growing studies exploring these concepts among indigenous populations and those living in the global South [26]. For example, a study on Thai ALHIV [27] found that spirituality and dignity played an essential role in maintaining mental wellness; in turn, this was associated with living responsibly and experiencing a better quality of life. There has also been an increase in instruments that measure singular mental wellness constructs (i.e., self-esteem) or general mental well-being [26,28,29,30]. Many of these instruments were developed with general adult and adolescent populations living in high-income countries (HICs) [26,28,29,30]. Due to the cost and time associated with instrument development, instruments developed in HICs are typically validated for use in other contexts, such as the KIDSCREEN measures or WHO-5 well-being index [31,32,33]. However, we need to be critical of how these instruments were validated and of the type of validity that was established. For example, the study by Balthip et al. indicates that spirituality is an essential indicator of mental wellness among Thai ALHIV. Nevertheless, this indicator is often not included in commonly used and validated measures such as the KIDSCREEN measure. This raises pertinent questions regarding the relevance of such instruments and what indicators of mental wellness should be targeted to improve outcomes among ALHIV in different contexts. Relatedly, the World Health Organization (WHO) published guidelines on mental health promotion and prevention interventions for adolescents in 2020 [21]. A leading recommendation of the report states that psychosocial interventions should be provided for ALHIV, as these are shown to promote positive mental health and reduce mental disorders [21]. However, the report also indicates that due to a lack of evidence, it was not possible to provide specific recommendations on psychosocial interventions to promote positive mental health, and that additional research is required to improve mental health trajectories. Throughout the report, mental-wellbeing and mental functioning were listed as positive mental health outcomes, while no specific indicators of positive mental health were stated [21]. 

As indicated, there is a limited understanding in terms of which mental wellness constructs would be most relevant to improving the health and well-being of ALHIV, especially in low- and -middle-income countries (LMICs) [21]. While there are mental wellness instruments that have been validated with adolescents in various contexts, and these mental wellness construct measures may be potentially relevant to ALHIV, we need to consider that mental wellness as a social construct is influenced by time, culture, and age. Therefore, the lived experiences of ALHIV shape their perception and understanding of mental wellness and its associated constructs [20]. For example, HIV is a highly stigmatized condition; therefore, approaches aimed at improving self-acceptance among ALHIV will not be the same as approaches used to improve self-acceptance among adolescents who are not living with HIV. From a health equity perspective, it is crucial that we develop a comprehensive understanding of how mental wellness is perceived by ALHIV to identify relevant indicators and develop appropriate instruments to measure mental wellness in this population in the South African context. Considering the increased conversation around defining concepts such as mental wellness and positive mental health, this is an opportune time to explore the meaning and conceptualization of these from an African perspective to ensure that cultural and indigenous views of health and wellbeing are prioritized rather than transmuting concepts from the West. 

## 2. The Mental Wellness Measure for Adolescents Living with HIV 

There is an increasing need for mental wellness measures for ALHIV to provide much-needed data on the context and impact of mental wellness outcomes which can then be targeted in interventions and service delivery [21,34]. To address this, we developed a Mental Wellness Measure for ALHIV (MWM-ALHIV) instrument in the South African context. The MWM-ALHIV was developed by first conceptualizing mental wellness for ALHIV through a systematic review of mental health instruments used in research with adolescents [26,28], a photovoice study with ALHIV accessing treatment at three public healthcare facilities in the Cape Metropole District in South Africa [35,36], and an integrative review of mental wellness concepts emerging from research done with ALHIV in Africa [37]. The findings from the systematic review indicated that there is a lack of mental wellness measures developed specifically for adolescents living with a chronic condition (such as HIV) in the African context, thereby proving support for the development of a new instrument [26,28]. Furthermore, through the photovoice study, participants were able to lead the conversation on what mental wellness means to them and what mental wellness factors are most salient in their lives, while the integrative review provided insight on the mental wellness concepts that are relevant in the African context, and how these are shaped by the cultural context [35,36,37]. From these findings, we developed the Salutogenic Model of Mental Wellness (SMoMW) (Figure 1) adapted from Antonovsky’s [38] Salutogenic Model of Health as a theoretical guide to develop the MWM-ALHIV.

Antonovsky originally developed the Salutgogenic Model to emphasize the conditions that lead to health rather than the determinants of health [38]. The Salutogenic Model was born out of observations that people experience various stressors in their daily lives resulting in tensions which, if left unresolved, develops into the health damaging condition of stress [38]. However, exposure to stressors does not necessarily lead to stress and illness. Rather, Antonovsky noted that people who have access to resources are better able to cope with and resolve tensions than people with little or no resources [38]. Through the model, Antonovsky highlights the importance of focusing on the interplay of stressors and life experiences to move an individual towards health [38]. Two key concepts of this model are the generalized resistance resources (GRR), referring to any factor or characteristic which can be used to facilitate tension management, and a sense of coherence (SOC), referring to an individual’s capacity to manage and overcome stressors [38]. Based on the Salutogenic paradigm, the SMoMW was developed following extensive reviews of the literature and participatory research with ALHIV accessing treatment in the Cape Metropole District in South Africa, which helped us to identify key mental wellness constructs that are relevant to ALHIV. Therefore, the SMoMW can be used to guide health promotion interventions, inform youth-friendly services, and facilitate research activities. According to the SMoMW, mental wellness is expressed as overall SOC, which comprises cognitive (comprehensibility), behavioral (manageability) and motivational (meaning) mental wellness factors. SOC, in turn, is influenced by a range of ecological factors including the life experiences of ALHIV, their access to resources, potential exposure to stressors, and their life situation. In this sense, ALHIV who have a high SOC are more likely to access and mobilize resistance resources, which will strengthen their SOC, leading to better mental wellness. The SMoMW can be used to understand how these factors influence the mental wellness of ALHIV in context, and what points of intervention would be most useful, as shown in Table 1. The MWM-ALHIV was developed to measure the SOC aspect of the model, highlighting the key mental wellness concepts identified by ALHIV as being salient to their experiences of living with HIV [39]. Due to the heterogeneity of the group, it would be challenging to develop a measure that takes into account all contextual aspects such as the GRRs or the potential life stressors. However, the strength of the model is that it allows for such factors to be considered when designing health promotion interventions or guiding research. Therefore, the MWM-ALHIV can be used in conjunction with the model to guide the collection of demographic data (culture, age, mode of infection etc.), select supplementary tools to measure other aspects (i.e., exposure to violence, adherence, treatment fatigue), or interpret findings.

Ensuring validity is a crucial step in instrument development to determine the extent to which a measure accurately captures what it is intended to measure [31,40,41]. Validity testing is usually done during the pilot phase of a study, using methods such as factor analysis or Item Response Theory (IRT) [31,40,41]. However, it is equally important to test for content and face validity. In a previous study, we established content validity by engaging with experts in a modified Delphi Study to determine how adequately the domains and items represent the measurement of mental wellness among ALHIV [39]. The Delphi Study participants endorsed the measure by providing consensus on the relevance and representation of the domains and items. In its present state, the MWM-ALHIV includes 113 items and measures mental wellness as an overall SOC represented through three domains and 11 sub-domains (Table 1).

As mental wellness is increasingly prioritized, we need to ensure that ALHIV are part of the conversation (especially those living in LMICs) and are included in efforts to improve their health and well-being. According to the WHO and UNAIDS, including ALHIV in the research process is a key priority [34]. Therefore, the next step is to establish the face validity and improve on the content validity of the MWM-ALHIV by engaging with the target population as the next group of experts [34]. To this end, we conducted cognitive interviews with a group of ALHIV to determine to what extent the items in the measure are appropriate, acceptable, sensible, and relevant for the intended users. According to UNAIDS, the participation and leadership of ALHIV are crucial at all stages of informing HIV programming, including the design, implementation, and monitoring and evaluation. The MWM-ALHIV reflects this commitment, as it was developed using a participatory photovoice method, allowing ALHIV to lead the conversation and express what aspects of mental wellness are important to them and what role it plays in their lives. The cognitive interviews followed as a logical next step to meaningfully engage with participants and collaborate with them to improve the MWM-ALHIV. Once finalized, the MWM-ALHIV can be used to implement, monitor, and evaluate youth-friendly services and HIV programming for ALHIV [21].

## 3. Methods

Cognitive interviewing is a technique used to evaluate the content and face validity and applicability of a survey or instrument [41,42,43]. The cognitive interview methodology was first established in the 1980s as a means to improve the validity of an instrument by understanding the cognitive processes involved in answering response items [41,42,43]. This technique is used during the preliminary stages of instrument development to gain insight into the participant’s cognitive processes when responding to ensure they understand the questions as intended [41,42,43]. There are two main approaches to cognitive interviewing, namely the ‘think-aloud’ and ‘verbal probing’ techniques [41,42,43]. The think-aloud method involves participants verbalizing their thought processes while responding to each item in the instrument, with the interviewer documenting the participant’s thought processes [41,42,43]. On the other hand, the verbal probing approach involves the interviewer asking a series of probing questions aimed at eliciting detailed information from the participants after they have responded to the items. While the think-aloud method is advantageous in reducing biased responses, asking participants to verbalize their thoughts while answering a question can be an unnatural and difficult practice for participants, resulting in a significant cognitive burden [43]. As such, the interviews were conducted according to the verbal probing approach outlined by Willis and Artino, as this was considered appropriate for the given sample of adolescents [43].

### 3.1. Participants and Procedures

The MWM-ALHIV is intended to measure mental wellness among ALHIV aged 14–19 years. As such, we recruited participants matching those criteria from a public healthcare facility in the Cape Metropole District of South Africa. The Cape Metropole District is located in the Western Cape Province, with reports indicating that 6.76% of people living with HIV reside in the province [15]. The healthcare facility is located in a ‘low-income’ community and provides free healthcare to members of the community and the surrounding areas. Additionally, to be included in the study, participants had to speak English as a first or second language. Due to the sensitive nature of the study, we first made contact with a doctor at the facility who was previously in charge of running the Youth Adherence Clubs. The doctor was given the relevant information with the study and asked to aid in the recruitment due to issues around disclosure. Those who were interested in participating were put in contact with the researcher and received information sheets as well as assent and consent forms for those younger than 18 years. Following this, the researcher set up a time and date to interview each participant at their convenience. All interviews were conducted at the public healthcare facility.

### 3.2. The Lexical Context in South Africa

Cognitive interviews are based on the assumption that we can use language (think-aloud and verbal probes) to tap into the cognitive processes of the participant and explain the way participants mentally process and respond to items in surveys or questionnaires [43].

As language plays a significant role in this process, it is salient to consider the multilingual context in South Africa. There are 11 official languages in South Africa, with IsiZulu identified as being spoken by the majority of the population (23%), followed by isiXhosa (16%), Afrikaans (13.5%), English (10%), Sesotho sa Leboa (9%), Setswana (8%), Sesotho (8%), Xitsonga (4.5%), siSwati (2.5%), Tshivenda (2.5%) and isiNdebele (2%) [44]. In the Western Cape province, where this study is based, the main languages are Afrikaans (49%), isiXhosa (24.7%) and English (20.3%) [44]. During the apartheid era, English and Afrikaans were identified as the official national languages of South Africa, while Indigenous languages were marginalized [44,45,46]. In addition to declaring the 11 official languages, other efforts aimed at redressing the language inequality include policies which state that South African children be taught in their mother tongue in the first three years of schooling, after which they are taught through an English or Afrikaans medium until their final year of high school [46].

In post-apartheid South Africa, Afrikaans continues to be used widely in the media and basic education system, with English dominating as the language of urban life and that predominantly used in the media, business, government, and basic and higher education systems [44,46]. Thus, despite being spoken as a home language by a minority of the population, English is used as a second language and a common language of communication in urban areas [44,46]. Furthermore, language in South Africa is fluid, with the majority of the population speaking more than two languages [44]. Census data from 2011 indicated that the average South African speaks between two and three languages. As such, South Africans are considered to be a ‘code-switching’ people, meaning that they may use more than one language during a conversation [44].

### 3.3. Data Collection

The interviews were carried out by a researcher who is experienced in qualitative research, has had previous training and experience conducting cognitive interviews with adolescents, and has experience doing research with ALHIV. The interviews were conducted in a private, quiet space to allow the participant to answer honestly. Before the start of each interview, the researcher reiterated the purpose of the study, and that the participant had the right to stop the interview process at any point if they no longer wished to continue. To reduce social desirability bias and to help participants feel comfortable, the researcher explained that the questions in the instruments were derived from photovoice interviews with other ALHIV who attended the healthcare facility in 2019. The interviews were conducted in December 2022. During the interview, the researcher sat next to the participant and read each question in the MWM-ALHIV aloud, along with the answer options, and then gave the participant the opportunity to select a response option. Following this, the researcher would ask probing questions based on the participant’s answer and their experience to assess the cognitive match between the intent of the question and the participant’s understanding and interpretation of the question. The probing questions included ‘why did you choose that answer?’, ‘what does [key term from questionnaire] mean to you?’, ‘can you explain what [key term] means to you in your own words?’, ‘what popped in your head when I said [key term]’, ‘I noticed you hesitated before answering that question, can you tell me more?’, ‘were any of the questions easy to answer?’, ‘were any of the questions hard to answer?’ These probes allowed participants the opportunity to reflect on their answers and to provide explanations to demonstrate their understanding. Additionally, the researcher asked questions to elicit macro-level engagement from the participants, which included ‘how would you ask [question from instrument] to your friends?’ or ‘what word [key term] would you and your friends use?’

As a psychologist and a woman of color who grew up in a post-apartheid South Africa, the researcher was aware of the power differences that could affect how participants interacted with her. She tried to minimize these by encouraging casual interaction with the participants, using local colloquialisms, and building a rapport with the help of the doctor. After each session the researcher had a debriefing session with the participant, giving them an opportunity to reflect on the experience and ask questions. Following this, the participants were given an incentive of ZAR 150 (USD 9) as a thank you.

### 3.4. Data Analysis

All interviews were audio recorded and transcribed. Additionally, the researcher made detailed field notes during and after the interviews to aid with the analysis. The transcripts were analyzed thematically to identify common themes and patterns that may indicate problems with the items, ambiguous wording, or potential sources of bias.

### 3.5. Ethics

Ethical clearance for this study was obtained by the University of the Western Cape. Each participant received and returned signed consent/assent forms before the start of the interviews. Participants who were younger than 18 years provided signed parental consent forms. Participants were reminded that they could withdraw from the study at any time without any negative consequences and that all of their information would be kept private and confidential.

## 4. Findings

We conducted interviews with nine participants accessing treatment at the public healthcare facility. The aim of the current study is to establish the face validity of the instrument rather than assess the mental wellness of the participants. Therefore, according to Willis [43] this sample size is deemed sufficient to confirm patient understandability of an item. Additionally, the analysis revealed no new themes emerging. The findings yielded enough information to identify key issues in the items to be revised. As cognitive interviewing is an iterative approach, it would be more useful to integrate the revisions and then conduct additional rounds.

The participant demographics are presented in Table 2. As shown in the table, the majority of participants (*n* = 7) indicated isiXhosa as their home language and English as their second language. As indicated earlier, this would mean that the seven Xhosa-speaking participants would receive their secondary education in English. The two Afrikaans participants were educated in their home language and study English as a second additional language, in line with the Department of Basic Education’s curriculum. All of the participants completed or are currently enrolled in school and are therefore considered to be literate. Furthermore, all participants were able to converse fluently in English, which, as mentioned, is a reflection of the urban setting. Two of the participants stated that they completed level 4 in the School of Skills—an alternative education institution for pupils who are unable to cope with or develop in mainstream institutions [47]. Pupils are enrolled at age 14 or 15 years and complete 4 years of schooling [47]. As Afrikaans in the predominantly spoken language in the Western Cape, it may be surprising that most of the participants in this sample spoke isiXhosa as their home language. However, this may also reflect the racial disparities and economic inequalities which drive the HIV epidemic in South Africa. Furthermore, all participants indicated that they were perinatally infected; thus, suggesting that the overrepresentation of Xhosa speaking participants in this study may be associated with the high HIV prevalence among Black South African women who lacked access to ART and the prevention of mother to child transmission (PMTCT) services (which was only initiated in 2002) [48].

The findings from this research highlight the value and importance of working with ALHIV; through engagement and feedback from the participants, we were able to work together to identify question failures and problems related to the face validity of the instrument which were not identified during the Delphi study.

These issues are classified into three themes, namely: comprehension mismatch, ‘big’ or difficult words, and sentence structure and question relevance (Table 3). These question failures should be understood in the context of the South African language and education landscape, and raise questions about survey development in South Africa. Generally, the older participants (18/19) were more confident in verbalizing their answers and thought processes than the younger adolescents. Both participants A01 and A02 attended a School of Skills; however, A01 spoke about her learning difficulties and demonstrated an awareness of her strengths and limitations. As such, she was able to clearly explain how she answered specific questions and engaged with the researcher on a macro level by making suggestions.

The younger adolescents (15/17) struggled to verbalize their thought processes. As one participant said, ‘I know what it means, I am just struggling to find the words to explain’. In these cases, the researcher would try to elicit a response by asking participants if they could provide examples or a similar word or suggested that they say it in their home language. For example, upon prompting, A02 said she would describe a valuable person as an ‘important’ person. In a similar study aimed at adapting a measure of grief among South African adolescents, it was noted that since cognitive interviewees are tasked with explaining how they experience and interpret specific words and phrases, polyglot contexts represent an exceedingly complex environment for research implementation [49]. This raises the question of whether participants’ struggles to verbalize their thoughts are related to a lack of understanding. Indeed, the relationship between language and cognition is a persistent question in scientific inquiry [43,49].

The participants’ responses to items were analyzed in light of this context. Given that all participants were bilingual and had different literacy levels, items were revised to reflect the lowest literacy level. In other words, an item was flagged for revision even if only one participant struggled to comprehend it.

## 5. Discussion

There is a growing recognition on the importance of improving mental wellness among ALHIV to support lifelong adherence to ART and ensure that they reach their full potential across the course of their lives. Strengthening mental wellness among ALHIV in South Africa is especially critical considering the high prevalence rates of HIV, the impact of the epidemic, and the health risks they are exposed to as a result of pervasive inequalities within the country. While it may take years to address the structural problems, focusing on mental wellness through mental health promotion interventions may offer protection from health risks to ALHIV and strengthen their capacity to thrive. However, in resource restrained contexts such as in South Africa, we need evidence-based responses to maximize the impact and outcome of interventions and services. To generate quality evidence, we require robust tools. Therefore, we set out to validate the MWM–ALHIV as a first step towards developing an age and culturally appropriate measure of mental wellness made for ALHIV, with ALHIV.

This study provides an example of how cognitive interviews with ALHIV in South Africa can be used to improve the face validity of the measuring instrument, MWM-ALHIV. The MWM-ALHIV was developed to address the gap in research on mental wellness among ALHIV. While there are numerous instruments aimed at measuring positive mental health, our goal was to develop an instrument that is culturally sensitive and captures the world view of ALHIV in the South African context. A strength of this measure is that it has been developed using participatory approaches with ALHIV that enabled the researchers to understand which aspects of mental wellness are important to them, how they talk about and understand it, and the role it plays in facilitating adherence to ART. The current study is concerned with improving the face validity of the instrument and represents a snapshot in the process towards establishing the psychometric properties of the instrument.

Similar to Taylor et al. [49], we found that the language issues which emerged from the interviews reflect the challenges of the South African landscape. Translating the questionnaire to Xhosa or Afrikaans is necessary to accommodate the larger population. However, this may be a challenging process. For example, when asked if the questions would be easier if they were translated to Xhosa, participant A01 suggested that it may not make a difference—even though she mainly speaks Xhosa at home, she also speaks other languages with her family and she never learned how to read or write in Xhosa. She further stated that due to her learning difficulties, it is easier for her to hear the questions verbally and then answer rather than reading them on her own. However, this may reflect a more complex problem in South Africa related to verbal and written language use. According to Chimbga and Meier, South African learners that were tested during an international comparative evaluation of reading literacy through the Progress in Reading International Literacy Study performed poorly, despite most writing in their home language [46].

Therefore, given the complex language issues, we agree with previous recommendations made by Taylor et al. [49] in that it may be advisable to adopt a multilingual approach in conducting cognitive interviews and even the survey design itself within the South African context. In other words, during the interviews, participants should be allowed to answer in their language of choice or to code-switch. Following this, decisions can be made regarding the appropriate language choices for the instrument [49]. To be representative of the fluid multilingual context and code-switching nature of South Africans, considerations should be made to have the instrument translated into a version which may include multiple languages or colloquialisms – such as Afrikaaps - in addition to a standard English version of the instrument or other translations [49]. Afrikaaps (also known as Kaaps) is a language created in settler colonial South Africa which developed as a result of encounters between indigenous African groups (Khoi and San) and slaves brought in from Southeast Asia and Portuguese, Dutch and English settlers [50]. In contemporary South Africa, the language is commonly spoken by working class speakers in the Cape Flats (an area in Cape Town where people were forcibly moved during the apartheid era) [50]. The language has been established since the 1500s and was first taught in madrassahs (Islamic schools). In later years it was appropriated by Afrikaner nationalists [50].

Additionally, we found that the interviews supported the appropriateness of the instrument and supported the rationale of the SMoMW. Therefore, even though certain domains in the model and instrument are believed to reflect Western concepts and values, such as self-esteem and self-acceptance, we found that participants resonated with and responded well to these categories and reflected the findings from our previous photovoice study, thereby demonstrating the confirmability of the findings. Furthermore, even though the mental wellness concepts originated from Western perspectives, we included and adapted these based on our engagement with ALHIV participants. It may be that these concepts perform well cross-culturally, as South Africa is considered a multi-cultural country; yet many of the systems and institutions are based on Western principles and values. While participants in this study were raised within collectivist cultures, they were also exposed to and expected to adapt to individualistic cultures and values that are perpetuated in their social circles. For example, when explaining an answer to ‘someone in my family accepts me’, participant A02 stated that it helped her feel like she was ‘just like’ her other family members and that she was ‘a normal person’. Therefore, the connectedness and acceptance she felt from her family members enabled (representing the collective values) her own self-acceptance and boosted her self-esteem (representing individualistic values) [35]. Similarly, A06 mentioned that living with HIV did not define who he is ‘because if it did, I would have killed myself’, indicating that self-acceptance is a motivator to continue living and receiving treatment.

On the other hand, participant A05 indicated that she ‘somewhat agrees’ with self-acceptance questions such as ‘I am kind to myself’, ‘I am happy with the way I am’, and ‘I am living with HIV, and I am okay with that’. When asked to explain her answers, the participant revealed that a few months prior she was would have answered ‘disagree’ to those questions, as she felt bad about herself as someone living with HIV. Consequently, this impacted her adherence to treatment. However, she described that she was on a journey, and she has started feeling more accepting of herself, but she is not ‘quite there yet’. The participant was one of the younger adolescents (15 years) and was informed of her HIV status when 11 years of age. Based on the SMoMW, and her answers to the MWM-ALHIV, we can hypothesise that even though she struggled with adherence, the connectedness she feels from her family and friends and the support she received from the doctor provided a buffer which helped her along her journey of self-acceptance, which in turn is giving her the tools she needs to re-commit to her treatment. Additionally, age is considered part of the life context in the SMoMW, which can play a role in mental wellness and health outcomes. For example, in comparison to the older participants, A05 (like her peers) is at the stage where she may be more impulsive, rebellious and/or forgetful when it comes to her treatment. During the interview, she mentioned that sometimes when she is out with her friends, she has so much fun that she forgets to take her medication, but when she does remember she would rather skip the dose if it is too late. On the other hand, older participants may have been through more life experiences which brought them further along the journey of self-acceptance. Additionally, they have been in mainstream adult care for a longer period of time, which would require them to manage their treatment independently.

The aforementioned indicates that the MWM-ALHIV has the potential to screen for mental wellness among ALHIV in a way that can indicate high levels of mental wellness, but can also pick up on areas where they may be struggling, without necessarily being in ‘crisis’. For example, the struggles A05 speaks about in her journey to adherence and maintaining self-acceptance may not have been flagged using instruments that are frequently used in assessing mental health among ALHIV such as the Child Depression Inventory (CDI) or the strengths and difficulties questionnaire [51]. However, her answers on the MWM-ALHIV suggest that she may benefit from additional support that specifically targets her self-acceptance and addresses issues around treatment reminders that are appropriate for her age.

In addition, findings from our systematic review of mental wellness instruments for adolescents indicated that there are relatively few instruments measuring mental wellness [often referred to as general mental well-being] (e.g., the Warwick-Edinburgh Mental Wellbeing Scale [52] and the Mental Health Continuum-Short Form [53]). Instead, the majority of the instruments measured singular indicators of mental wellness, such as connectedness (the Milwaukee Youth Belongingness Scale [54]), or self-esteem (e.g., the Rosenberg Self-esteem Scale [55]). Instruments measuring multiple indicators are preferred over instruments that focus on one mental wellness measure, because the former can provide a more comprehensive overview of both eudemonic and hedonic dimensions of mental wellness [56]. Various scholars have argued that both hedonic (feeling well) and eudemonic dimensions (functioning well) should be emphasized and included in measures to provide a more holistic view of adolescent mental wellness [57,58]. Particular strengths of the MWM-ALHIV are that the measure includes both general questions reflecting eudemonic and hedonic dimensions of mental wellness, as well as those specific to living with HIV. Furthermore, unlike the other instruments identified in the review [26,28] that originated in developed countries, the MWM–ALHIV was developed after extensive research to first conceptualize mental wellness for ALHIV in the African context. The MWM–ALHIV is the first mental wellness measure for ALHIV in the South African context that was developed in a South African setting.

## 6. Conclusions

This study was principally undertaken to determine the face validity and improve upon the content validity of the MWM–ALHIV. The MWM–ALHIV was developed as an age and culturally appropriate measure for ALHIV in the South African context. Unlike other instruments measuring general mental wellness or aspects of mental wellness that have been developed in the Western context and subsequently adapted to other cultures and contexts, the MWM–ALHIV was developed with ALHIV to ensure the appropriateness and relevance of the domains and to reflect their lived experiences. The cognitive interviews represent the next logical step in the study to include the voices of ALHIV in the instrument development process. Based on the responses from participants, revisions were made to improve the overall readability and comprehension of the measure. Additionally, the interviews also provide further insight into the appropriateness and confirmability of the domains of the measure and the SMoMW. Furthermore, the findings provide insight into considerations of language and implementation related to cognitive interviews and survey development with adolescents in the South African context. This study represents a snapshot of a larger project aimed at conceptualizing and developing a measure of mental wellness for ALHIV. Following the instrument development process, we aim to pilot the instrument and engage in further rounds of cognitive interviews to establish its psychometric properties.

## 7. Study Limitations and Recommendations

As a qualitative study, certain limitations are noted. However, due to the amount of rich data gathered from the interviews, we view this as a pilot stage which provided lessons learned for future rounds of cognitive interviews, which is in line with the cognitive interview iterative approach; we may conduct these rounds before the pilot testing of the instrument and after. The sample of participants in this study lived in an urban area and accessed treatment at a public healthcare facility from a specific community. We recommend further rounds of the cognitive tests be done with participants from other urban communities in addition to rural communities. Furthermore, based on the lessons learned, we would aim to recruit interviewers that speak isiXhosa so that participants may answer in the language of their choice. While the researcher offered the participants the opportunity to explain some concepts in their own language, she was limited in following up with further probing questions. As mentioned, to reflect the multilingual, fluid language of the South African landscape, we would recommend that this instrument be translated into appropriate languages and different dialects, or include some code-switching between languages, as participants may find this more relatable and easier to navigate.

## Figures and Tables

**Figure 1 ijerph-20-04061-f001:**
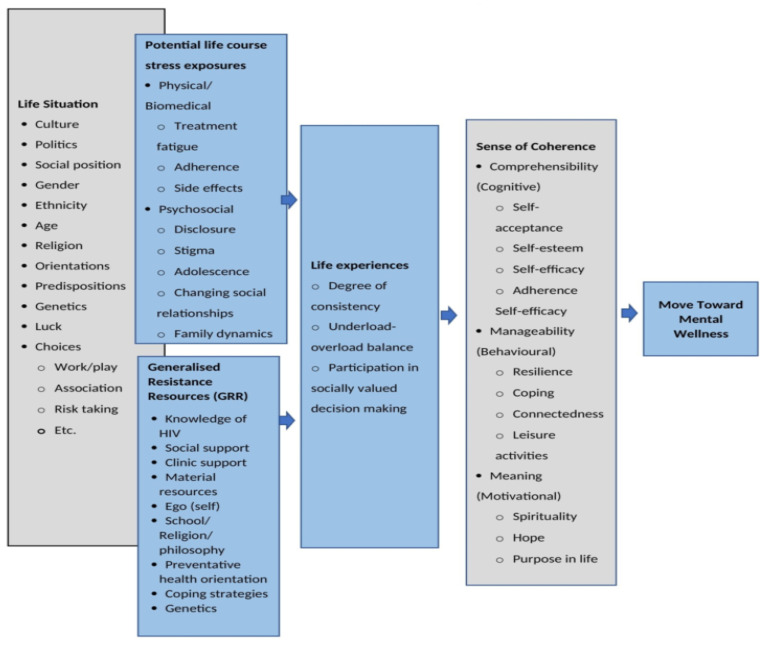
Salutogenic Model of Mental Wellness.

**Table 1 ijerph-20-04061-t001:** Mental wellness domains for the MWM-ALHIV.

Sense of Coherence	Sub-Domains (*n* = 11)	Number of Items (*n* = 113)
Comprehensibility (Cognitive)	Self-esteem	10
	Self-acceptance	9
	Self-efficacy	9
	Adherence Self-efficacy	13
Manageability (Behavioural)	Resilience	10
	Coping	10
	Connectedness	10
	Leisure activities	10
Meaning (Motivational)	Spirituality	7
	Hope	14
	Purpose in life	11

**Table 2 ijerph-20-04061-t002:** Participant characteristics.

Participant (*n* = 9)	Age	Gender	Home Language	Education Level
A01	19	Female	Xhosa	School of Skills (level 4)
A02	18	Other	Xhosa	School of Skills (level 4)
A03	18	Female	Xhosa	Gr. 12
A04	19	Female	Xhosa	Gr. 12
A05	15	Female	Afrikaans	Gr. 8
A06	19	Male	Xhosa	Gr. 12
A07	15	Male	Xhosa	Gr. 8
A08	17	Female	Xhosa	Gr. 10
A09	17	Male	Afrikaans	Gr. 10

**Table 3 ijerph-20-04061-t003:** Summary of Problem Types and Proposed Revisions.

Problem Type	Item	Domain	Explanation	Example	Proposed Revision
Comprehension mismatch	When I fail at something important to me, I remind myself that it is part of being human.	Self-acceptance	All of the participants struggled to answer this question and would often relate it to their experience of living with HIV	A02: ‘Sometimes I will feel lonely, because I’ll be like yes, I am HIV but that does not mean that’ A03: That I’m not going to die anytime soon, ‘I’m going to live long.’ A01: ‘Strongly agree, Yeah, I think I get really disappointed.’	Remove
	Even when I am bad at something, I still love myself.	Self-acceptance	Some of the participants interpreted ‘bad’ as doing something bad in character rather than failing at something	A03 ‘I love myself even when I am bad?...bad things like doing bad things’ A07: ‘I don’t love myself when I do bad things’	‘I love myself even when I fail at something’
	When something upsets me, it does not change how I feel about myself	Self-acceptance	All the participants understood the question, however one participant indicated that the wording may cause them to answer incorrectly	A04: ‘No, I strongly disagree…or is it agree?’	‘When something upsets me, I feel bad about myself’
	I am comfortable with who I am as a person	Self-acceptance	The question intends to determine if the participant accepts themselves, but the word ‘comfortable’ can have a different interpretation	A06: ‘Being in a relaxed atmosphere or something or just being your comfort zone that could be at home or our friends’	‘I love myself just the way I am’
	I am a valuable person, even if there are parts of myself that I do not like.	Self-acceptance	Some of the participants interpreted ‘parts’ to relate to the body rather than general aspects or characteristics	A06: ‘Body parts’ A01: ‘Like, any marks that you [gesturing to body]—like me, I have two [skin] colours… So, sometimes I look at my arms and look at my body…and I want to change it but I can’t’	‘I love myself, even if there are things about myself that I dislike’
	I do not have much to be proud of.	Self-esteem	Some participants interpreted the question to relate to certain material possessions that they may not have	A02: ‘I don’t have much to be proud of, but I know that I can take care of my health and some things.’ A06: ‘Like asset, type of things?’	‘I feel like a failure’
	I feel strong	Self-efficacy	Strong was interpreted differently by participants, with some relating it to physical strength	A01: ‘Because I don’t have a lot of energy as others because I have a heart problem, so I don’t have so much strength like others’ ‘Like, I mean, I feel like strong. It means that you are like you are strong. Like you have the ability to do anything.’	‘I have what it takes to succeed/achieve my goals’ or I have the ability to do anything I want in life’
	I find ways to take my treatment every day, even when I am around people who don’t know that I am living with HIV	Adherence self-efficacy	Most participants understood the question and responded with some of the ways they would take the treatment (e.g., go to the bathroom). However, one participant believed the question asked them to take the treatment in front of others	A06: ‘I never take treatment in front of everybody.’	‘When I am around people who don’t know my status, I will still try to find a way to take my treatment (e.g., go to the bathroom)’
	I know where to go in my community to get help	Resilience	Some participants indicated that the question may be vague. The question intends to assess whether participants know where to access resources in the community if they need them	A04: ‘Yeah, I would ask them. I don’t know. I have like, I’ve never been in the situation, like, to go and seek help from the community.’	‘If I had a problem, I would know where to go in my community to get help’
	I do things at school that make a positive difference (i.e., make things better)	Resilience	While some participants interpreted the question correctly, such as A03, others struggled to understand and answer the question (A01)	A03: ‘Making good friends, doing different things that are good…playing sports’ A01: ‘Something that’s like different from the other things that I do’	‘I work well with people my age’ ’I feel good when I am school’
	In general, I feel I am in control of my life		Some of the participants interpreted the question as referring to their ability to be independent rather than their ability to be in control of themselves, despite life challenges	A04: ‘I think being in control is being independent’ A08: ‘Disagree…I have to follow the rules at home’	‘I feel I am in control of myself’
Question relevance	I take my treatment every day, even when my eating habits have changed.	Adherence self-efficacy	The question did not resonate with participants	A06: ‘My eating habits never change.’ A08: ‘No, I don’t know how to answer this’	Remove
	I take my treatment every day, even when I have problems going to the clinic.	Adherence self-efficacy	The question did not resonate with participants	A06: ‘How would you take your treatment if you couldn’t go to the clinic?’	Remove
	I take my treatment every day, even when people close to me tell me not to	Adherence self-efficacy	The question did not resonate with participants	A04: ‘I never came across that’ A03: ‘No one told me’	Remove
	I take my treatment every day, even when I feel like people will judge me	Adherence self-efficacy	The question did not resonate with participants	A04: ‘Strongly disagree, I have never been judged by anyone’ A01: ‘No, people think the tablets are my for my heart’	Remove
	I know where to go for help when I have problems	Resilience	The wording of this question is similar to “I know where to go in my community when I need help. As such we proposed a revision.	A01: ‘I talk to my mom when I have problems’	‘When I have a problem, I can find what I need to solve it’
	Living with HIV has strengthened my faith or spiritual beliefs	Spirituality	The participants indicated the question was not very relevant	A04: ‘No…because, like, I feel like God has a purpose. Like God created everyone. There’s a reason why I’m living with HIV’ A06: ‘No it does not apply’	Remove
Big or difficult words and sentence structure	For me, life is about learning, changing, and growing despite my circumstances	Resilience	The wording of the question made it difficult for participants to understand	A01: ‘So, yeah, there’s a lot of challenges here outside.’ A09: ‘Strongly agree… I don’t know’	‘I see life challenges as a chance for me to learn and grow’
	I think it is important to have new experiences that challenge how you think about yourself and the world	Coping	The wording of the question made it difficult for participants to understand	A06: ‘Doing things you’ve never done before?’	Remove, similar as previous question
	When I have problems I think about different solutions to solve the problem	Coping	Most participants understood the question, but A01 offered a suggestion to make it easier	A01: ‘you can change it to make it easier’	‘When I have a problem, I think about different ways to solve the problem’
	I try to avoid difficult situations as much as possible	Coping	The wording of the question made it difficult for participants to understand	A01: ‘Like, those friends, there’s always that thing that you be talking and then there will be that one person who will and report that thing and it starts to be a big thing.’	‘I avoid my problems’
	I have friends I’m really close to and trust	Connectedness	Participants interpreted the question as having friends that they trust. Some have close friends, but they do not trust them with their status. The question was split into two to make it easier	A09: ‘Disagree, my friends don’t know’ A02: ‘I have friends, but I have not told them about my status’	‘I have friends I am really close to’ AND ‘I have friends that I trust’
	It is important to me that I feel satisfied by the activities that I take part in.	Leisure activities	The wording of the question made it difficult for participants to understand	A07: ‘Satisfied is…I don’t know how to explain it’ A08: ‘No, sometimes I just come home from school and do homework’	‘I have hobbies that make me feel happy’
	I will be able to provide for myself	Hope	Most participants understood the question, but A01 offered a suggestion to make it easier	A01: ‘Provide is a big English word. Maybe you can say ‘take care of myself’	‘I will be able to take care of myself’
	I will be able to provide for my family	Hope	Similar to previous		‘I will be able to take care of my family’
	I feel optimistic about the future		None of the participants understood the word ‘optimistic’	A04: ‘What does that mean?’ A01: ‘That is another big word’	‘I feel good about my future’
	I feel a sense of purpose in my life AND I have a sense of direction in life.	Purpose in life	These two questions were seen as similar, and some participants had difficulty understanding	A06: ‘it’s the same meaning’ A03: ‘sense of direction?’	‘I know where I want to go in life’
	I enjoy making plans for the future and working to make them come true		The question was seen as asking two different things. A04 recommended to split it into two questions	A04: ‘The question is in-between [difficult and easy to answer]’	‘I enjoy making plans for the future AND I work hard to achieve my goals’

## Data Availability

The data presented in this study are available on request from the corresponding author. The data are not publicly available due to the sensitive nature of the study and to protect the privacy of the participants.

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
