# Peer review of "Asking the Experts: Using Cognitive Interview Techniques to Explore the Face Validity of the Mental Wellness Measure for Adolescents Living with HIV"

_ijerph, 2023, doi:10.3390/ijerph20054061_

Round 1

Reviewer 1 Report

Dear Authors, 

Your topic is of great relevance to the psychological well-being of young people, especially those facing atypical developmental trajectories. Your manuscript is well-written and presents the theoretical background very well. 

However, it claims to be an original article and not a review article; therefore, the methodology part should be preponderant. Even in the context of qualitative research, the sample size is not acceptable to scientifically read the results; moreover, it is not accompanied by any other measures that could strengthen what is stated as a conclusion. Furthermore, as patients are included in a regular clinical course, additional primary clinical outcomes should be considered, as well as their clinical status compared in terms of therapy and treatment. 

I, therefore, believe that in these terms, the article cannot be published, and I suggest, considering the good theoretical background authors provide and the relevance of the subject, to present it as a review of the literature accompanied by an in-depth qualitative study in the field rather than as a validation of a scale for well-being used and usable in the clinical setting with patients.

Author Response

Reviewer 1

Dear reviewer, we thank you for taking the time to read and review our manuscript and providing feedback. We hope we have addressed the points you raised to your satisfaction. Please see the response below:

  • Your topic is of great relevance to the psychological well-being of young people, especially those facing atypical developmental trajectories. Your manuscript is well-written and presents the theoretical background very well.  However, it claims to be an original article and not a review article; therefore, the methodology part should be preponderant. Even in the context of qualitative research, the sample size is not acceptable to scientifically read the results; moreover, it is not accompanied by any other measures that could strengthen what is stated as a conclusion. Furthermore, as patients are included in a regular clinical course, additional primary clinical outcomes should be considered, as well as their clinical status compared in terms of therapy and treatment.  I, therefore, believe that in these terms, the article cannot be published, and I suggest, considering the good theoretical background authors provide and the relevance of the subject, to present it as a review of the literature accompanied by an in-depth qualitative study in the field rather than as a validation of a scale for well-being used and usable in the clinical setting with patients.
    • Thank you for your interest in this study and your notes. Cognitive interviewing is a popular method for evaluating survey questions and offers a detailed depiction of meanings and processes used by respondents to answer questions which impact the quality of the survey data. To address the concerns you raised, we added to the manuscript to mention that this study provides a snapshot look in a larger process of validating the proposed instrument. We also added justification for the sample size as Willis (2005) states that a sample of 7-10 is considered sufficient to confirm patient understanding. Indeed, we found that after the 9 interviews, no knew themes emerged in terms of identifying question failures. As such we added this in the manuscript to explain that it is more valuable at this stage to take the feedback from the participants into consideration and address the errors identified before engaging in further rounds of cognitive interviewing and piloting the instrument with a sample of ALHIV to establish the reliability and other forms of validity (i.e. criterion, construct etc). As mentioned in the manuscript, this study has been done as part of a larger study which included reviews, photovoice interviews and engaging with a panel of experts in a Delphi Study. Therefore, the purpose of the cognitive interviews in this study was to establish the face validity to improve the instrument and ready it for further rounds of testing. According to scholars like Devellis and Krause, doing cognitive interviews and delphi studies before piloting a tool is critical, yet often overlooked and not reported on. Given the unique nature of language in South Africa, and the importance of working with ALHIV, we believe that this paper provides insights and lessons learned and adds to the body of knowledge around the importance of doing cognitive interviews (with ALHIV in the South African context) before piloting an instrument. We hope that these notes and the edits we have made provide clarification and address the points you have raised. Thank you again for engaging with us on this and for helping us to improve the quality of the manuscript.

Reviewer 2 Report

The sentence requires quotation of a source:

"Relatedly, the WHO published Guidelines on mental health promotive and preventive interventions for adolescents in 2020"

Is it the source [25] that should be quoted?

You should find another way to quote the source for the sentence:

"However, it is equally important to test for con-118 tent and face validity. In a previous study, we established content validity by engaging 119 with experts in a Delphi Study to determine how adequately the domains and items rep-120 resent the measurement of mental wellness among ALHIV (submitted for publication)."

Please, check the rules for citing an unpublished manuscript.

You should check the results part of the manuscript for some linguistic errors related to sequence of tenses. 

You state that:

"Additionally, we found that the interviews supported the content validity of the instrument and supported the rational of the SMoMW."

Usually, when it is considered that content validity is supported, an index of content validity is reported.

The sentence requires quotation of several sources:

"Additionally, findings from our systematic review of mental wellness instruments for adolescents indicated that there are relatively few instruments measuring mental wellness [often referred to as general mental well-being] (e.g. Warwick-Edinburgh Mental Wellbeing Scale, Mental Health Continuum-Short Form)."

You should quote the source of systematic review and the source for each scale.

You should quote some sources for the sentence:

"Instead, the majority of the instruments measured singular indicators of mental wellness such as connectedness (Hemingway Measure of Adolescent Connectedness, Milwaukee Youth Belongingness Scale) or self-esteem (e.g. Rosenberg Self-esteem Scale, The Self-esteem Questionnaire)."

You should quote the source for each scale.

You should separate the section Findings and Discussion into two sections. 

You should clearly state and summarize your findings regarding face validity of the instrument in Discussion section.

Author Response

Reviewer 2

Dear Reviewer, we thank you for taking the time to read and review our manuscript and for providing feedback. We hope we have addressed the points you raised to your satisfaction. Please see the response below

  • "Relatedly, the WHO published Guidelines on mental health promotive and preventive interventions for adolescents in 2020" Is it the source [25] that should be quoted?
    • Yes, the [25] reference was the source. We have included the source after the sentence.

  • You should find another way to quote the source for the sentence: "However, it is equally important to test for con-118 tent and face validity. In a previous study, we established content validity by engaging 119 with experts in a Delphi Study to determine how adequately the domains and items rep-120 resent the measurement of mental wellness among ALHIV (submitted for publication)." Please, check the rules for citing an unpublished manuscript.
    • Thank you, we have amended the citation accordingly. 

  • You should check the results part of the manuscript for some linguistic errors related to sequence of tenses. 
    • Thank you for noting this, we have proofread the paper and have fixed the linguistic errors.

  • You state that: "Additionally, we found that the interviews supported the content validity of the instrument and supported the rational of the SMoMW." Usually, when it is considered that content validity is supported, an index of content validity is reported.

  • Thank you for this comment, to clarify, the initial aim of this study was not to establish the content validity of the SMoMW (the theoretical model), rather we aimed to establish and improve the face validity of the Mental Wellness Measure for Adolescents Living with HIV (MWM-ALHIV) by conducting cognitive interviews with a sample of ALHIV to explore how they understand and respond to the items in the measure. However, through the interviews, participants explained their answers in a way that reflected the findings from our previous studies which we used to develop the SMoMW. As such, we tried to explain that the model is holding up and confirming the findings from the previous work. We sought to establish the content validity of the MWM in a previous study by conducting a delphi study with experts in the field to reach consensus. I have tried to change the wording in the manuscript to reflect this purpose and to provide clarity.

  • The sentence requires quotation of several sources: "Additionally, findings from our systematic review of mental wellness instruments for adolescents indicated that there are relatively few instruments measuring mental wellness [often referred to as general mental well-being] (e.g. Warwick-Edinburgh Mental Wellbeing Scale, Mental Health Continuum-Short Form)."You should quote the source of systematic review and the source for each scale.
    • Thank you for noting this, I have added the sources 

  • You should quote some sources for the sentence: "Instead, the majority of the instruments measured singular indicators of mental wellness such as connectedness (Hemingway Measure of Adolescent Connectedness, Milwaukee Youth Belongingness Scale) or self-esteem (e.g. Rosenberg Self-esteem Scale, The Self-esteem Questionnaire)."You should quote the source for each scale.
    • Thank you for noting this, I have added the sources 

  • You should separate the section Findings and Discussion into two sections. 
    • Thank you for this recommendation, we have separated the two sections

  • You should clearly state and summarize your findings regarding face validity of the instrument in Discussion section.
    • Thank you, as we have separated the findings and discussion, we have provided more detail in the discussion section

We hope that these notes and the edits we have made provide clarification and address the points you have raised. Thank you again for engaging with us on this and for helping us to improve the quality of the manuscript.

Reviewer 3 Report

The manuscript titled: Asking the experts: Using cognitive interview techniques to explore the face validity of the Mental Wellness Measure for Adolescents Living with
HIV highlights the relevance of improving the mental health among adolescents living with HIV in the South African context. This paper also addresses the relevance of developing instruments that properly measure mental health constructs using qualitative research. Cognitive interviewing was conducted to evaluate the content, face validity and survey implementation of the MWM-ALHIV instrument with 9 ALHIV participants. The paper is well written and structured. A strength of this paper is that addresses the relevance of mental health promotion focusing on positive mental health outcomes. However, this paper would need to clarify and strength the arguments on how this paper contributes to the wellbeing of South African adolescents and mental health interventions. Furthermore, the paper needs a more detailed explanation on how the MVM incorporates the conceptualization of wellbeing from a cultural perspective with the population of interest. Below, I suggest areas to improve this paper:

1.Introduction: The introductory section is appropriate conducting a good literature review. However, the literature review seems to lack more articles addressing the context and group of interest to contextualize HIV in South Africa. For instance incorporating papers that describe poverty levels, access to care and other structural challenges such as racism and discrimination. Mental health and wellbeing is impacted by those ecological factors that are not described in this paper.

On line 59 “a growing body of positive psychology” seems vague. The authors should explain how positive psychology applies to the South African context. This is a very Eurocentric/Western/ethnocentric theory that is difficult to applied to opressed populations or other countries when considering the historical and sociopolitical characteristics as well cultural differences in those particular contexts.

On line 91 the paper briefly touchs upon linguistic characteristics but the paper will benefit from more literature addressing the ecological and cultural context of the adolescents. For instance: who are more infected with HIV, cultural variations. Further, most of the participants belong to ethnic groups with a collectivistic world view and in consequence with a very different conceptualization of mental health than Westernized standards. This should be address on the paper.

Line 367 Figure 1 is not explained and seems misplaced. It should be placed after line 111 where the authors describe that this was a theoretical guide. It will also relevant to explain the rational to only incorporate Sense of cohesion from the Salugtogenic Model and no other very relevant dimensions such as Life situation and GRR. The construct of wellbeing needs to be defined.

Line 112, I suggest to explain the model rational to use particular dimensions focusing on the self without considering the contextual and cultural factors contributing to the wellbeing in this case with the population of interest.

2. Participants: I suggest describing participant’s literacy level, age differences. The paper will benefit from a section explaining the characteristics on the different ethnics groups in South Africa and the contextualization of HIV. For instance, there is an overrepresentation of certain groups infected of HIV and this is not explained in the paper.

3. Data collection:  positionality/social and ethnic background of the researcher is not described, which may influence results, for example how the researcher was perceived by the adolescents. How the participants felt about this process, did they had an opportunity to give feedback on the research process, did they receive an incentive?.

4. Findings and conclusions: The authors should strength their arguments of why the instrument needs to be validated in the particular context of South Africa and case how this instrument is taking into consideration the cultural context of the participants. The authors should also explain on how the different domains of the MWM_ALHIV incorporated specifically the sub domain: self-esteem, self-acceptance, self-efficacy that are psychological constructs based in Westernized visions on mental health. On the contrary, the results indicate for instance the importance of family and how this boosted the self-esteem of the participants. These findings seem to reflect cultural values on family of collectivistic cultures. I suggest expanding on how these findings on family, collectivism informs the development and validation of the survey/instrument. In addition results will benefit from describing how the survey items were reevaluated after conducting the interviews and how those results informed the survey validation for instance which items seems to be more relevant or better measure positive mental health.

Line 377 this sentence seems of when describing limitations around generalization. This is not applicable to qualitative research. The goal of qualitative studies is not to generalize but rather to contextualize.

6. Conclusion. The conclusions are too short, the paper will benefit from expanding it with an academic discussion where the findings of the study are put into a conversation with existing studies in the field and how the findings specifically contribute to wellbeing as indicated in the previous sections.

Author Response

Reviewer 3

Dear Reviewer, we thank you for taking the time to read and review our manuscript and for providing feedback. We hope we have addressed the points you raised to your satisfaction. Please see the response below

  • 1.Introduction: The introductory section is appropriate conducting a good literature review. However, the literature review seems to lack more articles addressing the context and group of interest to contextualize HIV in South Africa. For instance incorporating papers that describe poverty levels, access to care and other structural challenges such as racism and discrimination. Mental health and wellbeing is impacted by those ecological factors that are not described in this paper.
    • Thank you, we have added more literature in the introduction section to provide context of the HIV epidemic in SA and other drivers of health which may impact the mental health of ALHIV. Please see line 46-79

  • On line 59 “a growing body of positive psychology” seems vague. The authors should explain how positive psychology applies to the South African context. This is a very Eurocentric/Western/ethnocentric theory that is difficult to applied to opressed populations or other countries when considering the historical and sociopolitical characteristics as well cultural differences in those particular contexts.
    • Thank you for this comment, we agree that it is important to consider the relevance of positive psychology as most of the work as been situation in Western context and adapted elsewhere. Therefore we need to be conscious of how these concepts are adapted or experienced in other contexts to ensure certain dimensions and nuances are not lost. We added information on in line 93-106 to clarify.

  • On line 91 the paper briefly touchs upon linguistic characteristics but the paper will benefit from more literature addressing the ecological and cultural context of the adolescents. For instance: who are more infected with HIV, cultural variations. Further, most of the participants belong to ethnic groups with a collectivistic world view and in consequence with a very different conceptualization of mental health than Westernized standards. This should be address on the paper.
    • Thank you, we added this information in the introduction section to match the South African context, as well as the findings and discussion section. As this paper specifically was aimed at establishing the face validity of the instrument, we did not ask participants to identify their race/ethnicity, rather is was more useful to focus on language. While we can assume that participants who identify Xhosa are Black African, one participant stated that her home language is Xhosa, but she also speaks her father’s language (indigenous to Namibia). Therefore, we tried to be careful about making race based assumptions. The inequalities related to the HIV epidemic and how these impact mental health are noted, but our main aim was to establish whether participants were able to understand the questions asked as intended and if they could relate to the questions. As we developed this measure using photovoice techniques, we found that participants were able to relate to the content.
  • Line 367 Figure 1 is not explained and seems misplaced. It should be placed after line 111 where the authors describe that this was a theoretical guide. It will also relevant to explain the rational to only incorporate Sense of cohesion from the Salugtogenic Model and no other very relevant dimensions such as Life situation and GRR. The construct of wellbeing needs to be defined. Line 112, I suggest to explain the model rational to use particular dimensions focusing on the self without considering the contextual and cultural factors contributing to the wellbeing in this case with the population of interest.
    • Thank you for noting this, we have provided more information and expanded on the explanation of the model line 169-203.
  • 2. Participants: I suggest describing participant’s literacy level, age differences. The paper will benefit from a section explaining the characteristics on the different ethnics groups in South Africa and the contextualization of HIV. For instance, there is an overrepresentation of certain groups infected of HIV and this is not explained in the paper.

As mentioned in the previous comment, we tried to add more information on representation to add context without making assumptions. We added information for context in the findings section and also referring back to the introduction section.

  • 3. Data collection:  positionality/social and ethnic background of the researcher is not described, which may influence results, for example how the researcher was perceived by the adolescents. How the participants felt about this process, did they had an opportunity to give feedback on the research process, did they receive an incentive?.
    • Thank you for this important point, we have included the iformation added line 234-230

  • 4. Findings and conclusions: The authors should strength their arguments of why the instrument needs to be validated in the particular context of South Africa and case how this instrument is taking into consideration the cultural context of the participants. The authors should also explain on how the different domains of the MWM_ALHIV incorporated specifically the sub domain: self-esteem, self-acceptance, self-efficacy that are psychological constructs based in Westernized visions on mental health. On the contrary, the results indicate for instance the importance of family and how this boosted the self-esteem of the participants. These findings seem to reflect cultural values on family of collectivistic cultures. I suggest expanding on how these findings on family, collectivism informs the development and validation of the survey/instrument. In addition results will benefit from describing how the survey items were reevaluated after conducting the interviews and how those results informed the survey validation for instance which items seems to be more relevant or better measure positive mental health.
    • Thank you, we expanded on the discussion section after separating it from the results section (also in relation to the information added in the introduction). As mentioned, we have developed the MWM-ALHIV and are in the process of establishing the psychometric properties. Before piloting the instrument, it is necessary to first establish the content validity (which we did through a delphi study) and face validity (current study). As mentioned in the paper, there is a need for valid mental wellness measures to provide evidence informing health promotion interventions and services. Currently, many of the mental wellness measures are those that have been developed in Western contexts and adapted elsewhere. One point mentioned in the introduction section is the recognition that while these measures demonstrate validity and reliability in other cultural contexts, it may be that what is being measured does not truly reflect the world view and lived experience of adolescents in these cultures, much less ALHIV. (i.e. unlike their peers, ALHIV experience self-acceptance is influenced by living with HIV). Therefore, the MWM-ALHIV was developed with and for ALHIV in South Africa. Validating the instrument in this context is necessary so that it may be used to generate evidence and improve mental wellness services for ALHIV.

  • Line 377 this sentence seems of when describing limitations around generalization. This is not applicable to qualitative research. The goal of qualitative studies is not to generalize but rather to contextualize.
    • The line has been removed and the limitation section has been expanded on

  • 6. Conclusion. The conclusions are too short, the paper will benefit from expanding it with an academic discussion where the findings of the study are put into a conversation with existing studies in the field and how the findings specifically contribute to wellbeing as indicated in the previous sections.
    • Thank you, we have worked to expand on the conclusion

Many thanks again for providing valuable feedback and insight to help us improve upon the manuscript. We hope that these notes and the edits we have made provide clarification and address the points you have raised.

Round 2

Reviewer 1 Report

changes to the manuscript help the readers in clarifying which is the context in which your study is placed. Even if you provided the appreciated reference for the reduced sample size I still see the problem in it and for this reason I suggest to add this to the study's limitations together with the missing association with patients' clinical treatments, which would be useful to better interprete their psychological wellbeing level that rise up from the interview.

Author Response

Dear reviewer 

Thank you for your time and providing feedback. We are happy to note that the context of the paper has been improved. To address your last point, we tried to clarify that the aim of the current study is not to assess or interpret the psychological wellbeing, but rather to ensure that the participants understand the questions being asked as intended. We added "The aim of the current study is to establish the face validity of the instrument rather than assess the mental wellness of the participants Therefore, according to Willis [43] this sample size is deemed sufficient to confirm patient understandability of an item." (line 353-353). We used the information from the cognitive interviews to address problems in the instrument. We mention in the manuscript that we aim to pilot the instrument to establish the psychometric properties, which we can then use on a larger sample. We hope that we have addressed your comments sufficiently.